# Post-traumatic stress disorder and associated factors among internally displaced persons in Africa: A systematic review and meta-analysis

**Amensisa Hailu Tesfaye**[1]*, **Ashenafi Kibret Sendekie**[2], **Gebisa Guyasa Kabito**[1], **Garedew Tadege Engdaw**[1], **Girum Shibeshi Argaw**[3], **Belay Desye**[4], **Abiy Ayele Angelo**[5], **Fantu Mamo Aragaw**[6], **Giziew Abere**[1]

1 Department of Environmental and Occupational Health and Safety, Institute of Public Health, College of Medicine and Health Sciences, University of Gondar, Gondar, Ethiopia, 2 Department of Clinical Pharmacy, School of Pharmacy, College of Medicine and Health Sciences, University of Gondar, Gondar, Ethiopia, 3 Department of Nursing, College of Medicine and Health Sciences, Jigjiga University, Jigjiga, Ethiopia, 4 Department of Environmental Health, College of Medicine and Health Sciences, Wollo University, Dessie, Ethiopia, 5 Department of Immunology and Molecular Biology, School of Biomedical and Laboratory Sciences, College of Medicine and Health Science, University of Gondar, Gondar, Ethiopia, 6 Department of Epidemiology and Biostatistics, Institute of Public Health, College of Medicine and Health Sciences, University of Gondar, Gondar, Ethiopia

* amensisahailu@gmail.com

**Data Availability Statement:** All relevant data are within the manuscript and its Supporting information files (S3 File).

## Abstract

### Background

Internally displaced people (IDPs), uprooted by conflict, violence, or disaster, struggle with the trauma of violence, loss, and displacement, making them significantly more vulnerable to post-traumatic stress disorder (PTSD). Therefore, we conducted a systematic review and meta-analysis to assess the prevalence and associated factors of PTSD among IDPs in Africa.

### Methods

A comprehensive search of electronic databases was conducted to identify relevant studies published between 2008 and 2023. The search included electronic databases such as PubMed, CABI, EMBASE, SCOPUS, CINHAL, and AJOL, as well as other search sources. The Preferred Reporting Items for Systematic Reviews and Meta-Analyses (PRISMA) guidelines were followed. Data were extracted using Microsoft Excel, and analysis was performed using STATA 17 software. The quality of the included studies was assessed using the JBI quality appraisal tool. A random-effects model was used to estimate the pooled prevalence of PTSD and its associated factors. The funnel plot and Egger's regression test were used to assess publication bias, and $I^2$ test statistics was used to assess heterogeneity. The protocol for this review has been registered with PROSPERO (ID: CRD42023428027).

### Results

A total of 14 studies with a total of 7,590 participants met the inclusion criteria. The pooled prevalence of PTSD among IDPs in Africa was 51% (95% CI: 38.-64). Female gender (OR

**Funding:** The author(s) received no specific funding for this work.

**Competing interests:** The authors have declared that no competing interests exist.

= 1.99, 95% CI: 1.65–2.32), no longer married (OR = 1.93, 95% CI: 1.43–2.43), unemployment (OR = 1.92, 95% CI: 1.17–2.67), being injured (OR = 1.94, 95% CI: 1.50–1.50), number of traumatic events experienced [4-7(OR = 2.09, 95% CI: 1.16–3.01), 8–11 (OR = 2.09, 95% CI: 2.18–4.12), 12–16 (OR = 5.37, 95% CI: 2.61–8.12)], illness without medical care (OR = 1.92, 95% CI: 1.41–2.29), being depressed (OR = 2.97, 95% CI: 2.07–3.86), and frequency of displacement more than once (OR = 2.13, 95% CI: 1.41–2.85) were significantly associated with an increased risk of PTSD.

## Conclusions

The findings of this systematic review and meta-analysis highlight the alarming prevalence of PTSD among IDPs in Africa. Female gender, marital status, number of traumatic events, ill health without medical care, depression, and frequency of displacement were identified as significant risk factors for PTSD. Effective interventions and the development of tailored mental health programs are needed to prevent PTSD among IDPs, focusing on the identified risk factors.

## Introduction

Internally displaced persons (IDPs) are individuals forced to flee their homes due to conflict, natural disasters, or other man-made or natural events [1]. These individuals face numerous challenges, including physical and psychological trauma, witnessing violence, loss of livelihoods, and separation from family and friends [2,3]. These challenges can contribute to the development of post-traumatic stress disorder (PTSD), a common mental health condition characterized by flashbacks, nightmares, hypervigilance, avoidance, and emotional numbness [4–6].

PTSD can have a profound and debilitating impact on the lives of IDPs, posing significant challenges to their recovery and reintegration into society [3]. The impact of PTSD on IDPs extends beyond the individual, affecting their families and communities as well. The emotional turmoil and behavioral changes associated with PTSD can strain relationships within families, leading to conflict, neglect, and further emotional distress [1,7]. PTSD can also lead to secondary problems such as substance abuse, self-harm, and suicide [8,9]. Moreover, the inability of IDPs with PTSD to contribute fully to their communities can hinder collective recovery efforts and exacerbate existing social and economic vulnerabilities [1,7,10]. The impact could have been highly significant in Africa because of several factors, including insufficient mental health services, the stigma associated with mental health problems, and the logistical challenges of providing mental health services to IDPs who are often living in remote areas [11–13].

Globally, more than 71 million people are internally displaced as of the end of 2022 across 120 countries because of conflict, violence, and disasters. This number shows an increase of 20% from the previous year [14]. Human and natural disasters that could potentially cause IDPs have been prominently reported in sub-Saharan Africa [15]. In 2020, Africa accounted for almost 40% of all new internal displacements globally, with natural disasters being the primary cause of displacement in 32 out of 54 African countries [16]. According to the United Nations Human Rights Commission (UNHCR), 42% of all IDP people globally have lived in Africa [17].

The prevalence of PTSD among IDPs varies widely, ranging from 3% to 88% depending on the specific country and population studied [18,19]. The prevalence of PTSD in East Africa

ranges from 11% to 80.2% [20–22]. Similarly, a meta-analysis study conducted in sub-Saharan African countries reveals that the magnitude of PTSD ranges from 12.3% in Central Sudan to 85.5% in Nigeria, and the majority of them reported to have more than 50% of the magnitude of PTSD [23]. This suggests that PTSD is a significant public health problem among IDPs in Africa.

Several factors have been identified as being associated with an increased risk of PTSD among IDPs. These factors include: female gender, young age, trauma, experiencing or witnessing violence, depression, anxiety, stress, low level of educational status, lack of social support, and economic hardship [20,23,24]. Beyond the previously mentioned factors, a number of other factors may also increase the likelihood of PTSD in IDPs in Africa. These factors include political instability and ongoing conflict, which can prolong the trauma and displacement cycle and make it more challenging for IDPs to find stability and security. This ongoing exposure to stress and uncertainty can exacerbate PTSD symptoms and hinder recovery efforts. Poverty and food insecurity are also common among IDPs in Africa, creating additional stressors and challenges. These socio-economic factors can contribute to feelings of hopelessness, despair, and a sense of being trapped in a difficult situation, further exacerbating PTSD among these vulnerable groups [25–28].

Despite the high prevalence of PTSD among IDPs in different African countries, there is a lack of comprehensive studies that show the pooled impact of PTSD and the nature of risk factors. Thus, a comprehensive study that can address the overall public health impact of internal displacement in terms of causing PTSD could be important to provide mental health services for IDPs. Therefore, the purpose of this systematic review and meta-analysis is to synthesize existing evidence on the prevalence of PTSD and its associated factors among IDPs in Africa. This information will be used to inform the development of interventions to prevent and treat PTSD among IDPs in Africa.

## Methods and materials

### Study setting

The study provides a comprehensive synthesis of existing research on PTSD prevalence rates and examines risk factors contributing to the development of PTSD among IDPs in African countries. According to the African Development Bank, there are 54 countries in Africa today [29].

### Protocol and registration

The protocol for this review was registered in the International Prospective Register of Systematic Reviews (PROSPERO), the University of York Centre for Reviews and Dissemination (Record ID: CRD42023428027, May 31st, 2023).

### Data sources and search strategy

This review and meta-analysis were conducted according to the guidelines of Preferred Reporting Items for Systematic Reviews and Meta-Analyses (PRISMA). The three phases drawn from the PRISMA flowchart were documented in the results to show the study selection process from identification to the included studies [30]. The PRISMA checklist was also used in the reporting of the systematic review and meta-analysis (S1 File).

We extensively searched articles on PubMed/MEDLINE, CABI, EMBASE, SCOPUS, CINHAL, and African Journal Online (AJOL) up to June 11, 2023. According to the African Development Bank, there are 54 countries in Africa [29]. The search terms were selected to capture

relevant articles on PTSD and IDPs in African countries. The search was conducted using a combination of keywords and controlled vocabulary (MeSH terms). The search strategy was adapted for each database as per their specific syntax and indexing terms. For the PubMed search, the following key terms were used in combination with the Boolean operators "AND" and "OR". ("Post-traumatic stress disorder" [All Fields] OR "Posttraumatic stress disorder" [All Fields] OR "PTSD" [All Fields]) AND ("Internally displaced persons" [All Fields] OR "Internally displaced people"[All Fields] OR "Internally displaced individuals" [All Fields] OR "IDP" [All Fields]) AND ("Associated factors" [All Fields] OR "Determinant factors" [All Fields] OR "Risk factors" [All Fields]) AND "Africa" [All Fields].

In addition to these electronic database searches, we searched the grey literature using website searches such as BioMed Central and National Institute of Mental Health, Behavioral and Brain Sciences—Cambridge University Press, etc., Google Search, and Google Scholar. We also searched the reference lists (bibliographies) of the included studies for additional relevant studies.

## Eligibility criteria

**Inclusion criteria.** Articles that met the following criteria were considered for inclusion in this systematic review and meta-analysis.

1. Population: internally displaced persons (IDPs).

2. Outcomes: articles reported the quantitative outcome of the prevalence of PTSD and associated factors among IDPs in Africa.

3. Study design: a cross-sectional study.

4. Study setting: studies conducted in African countries.

5. Publication issue: peer-reviewed journal articles published before 11 June 2023.

**Exclusion criteria.** Systematic reviews, qualitative studies, letters to editors, short communications, and commentaries were excluded. In addition, articles that were not fully accessible after three personal email contacts with the corresponding author and articles that did not indicate the outcome interest of this study were all excluded.

## Study selection process

The Endnote X9.2 (Thomson Reuters, Philadelphia, PA, USA) software reference manager was used to collect and organize search results and to remove duplicate articles. Three investigators (AHT, FMA, and GA) independently screened articles by their title, abstract, and full text to identify eligible articles using predetermined inclusion and exclusion criteria. The screened articles were then compiled together by three investigators (AHT, GSA, and AKS), and the disagreement between authors that arises during data abstraction and selection is solved based on evidence-based discussion and the involvement of other investigators (AKS, GTE, and AAA).

## Data extraction and management

Data were extracted using the Joanna Briggs Institute (JBI) data extraction checklist. Four review authors (AHT, FMA, GGK, and BD) extracted the data independently using a Microsoft Excel spreadsheet. The data extraction format included (name of first author, publication year, study country, study design, sample size, response rate, prevalence of PTSD, total number

of participants, factors associated with PTSD with their respective OR with 95% CI, and risk of bias). Disagreements between the review authors were resolved by a review by the other review authors based on an evidence-based discussion.

## Quality assessment of the studies

The quality of the included articles was assessed using the Joanna Briggs Institute (JBI) quality appraisal tools for analytical cross-sectional studies [31]. Three investigators (AHT, AKS, and GA) independently assessed the quality of the included articles. The assessment tool contains eight criteria: (1) clear inclusion and exclusion criteria; (2) description of the study subject and study setting; (3) use of a valid and reliable method to measure the exposure; (4) standard criteria used for measurement of the condition; (5) identification of confounding factors; (6) development of strategies to deal with confounding factors; (7) use of a valid and reliable method to measure the outcomes; and (8) use of appropriate statistical analysis. The risks for biases were classified as low (total score, 6 to 8), moderate (total score, 3 or 5), or high (total score, 0 to 2) (S2 File). Finally, articles with low and moderate biases were considered in this review. Disagreements that arose during the full-text quality assessment were resolved through evidence-based discussion with the involvement of other review authors (GGK, GA, and BD).

## Outcome of interest

The primary outcome of this review was the pooled prevalence of post-traumatic stress disorder (PTSD). It was reported as a percentage (%). The second outcome of this review was the pooled measure of the association between PTSD and associated factors among IDPs in Africa. It was determined using the pooled odds ratio (OR) with a 95% confidence interval.

## Statistical methods and data analysis

The extracted data were exported from a Microsoft Excel spreadsheet to STATA version 17 for further analysis. Heterogeneity among the included studies was quantitatively measured by the index of heterogeneity ($I^2$ statistics), in which 25%-51%, 50%-75%, and>75% represented low, moderate, and high heterogeneity, respectively [32]. The overall pooled estimate of PTSD among IDPs in Africa was computed using the metaprop STATA command. A subgroup analysis was conducted by a study country to see the difference in the pooled prevalence of PTSD between countries. The influence of a single study on the overall pooled estimate was assessed using a sensitivity analysis. Furthermore, the small-study effect was evaluated using the funnel plot test and Egger's regression test, with a p-value <0.05 as a cutoff point to declare the presence of publication bias. A p-value <0.05 was used to declare factors associated with PTSD to be statistically significant with a pooled odds ratio (OR) at the 95% confidence level. The results were presented using graphs, tables, text, and a forest plot.

## Results

### Searching process

A total of 4622 articles were identified using electronic databases and manual searching. After removing duplicate records, 3427 records were screened for this review. Based on their titles and abstracts, 3351 articles were excluded. In addition, 62 articles were excluded based on the exclusion criteria. Finally, a total of 14 articles were included in this review. The PRISMA flow diagram was used to summarize the selection process (Fig 1).

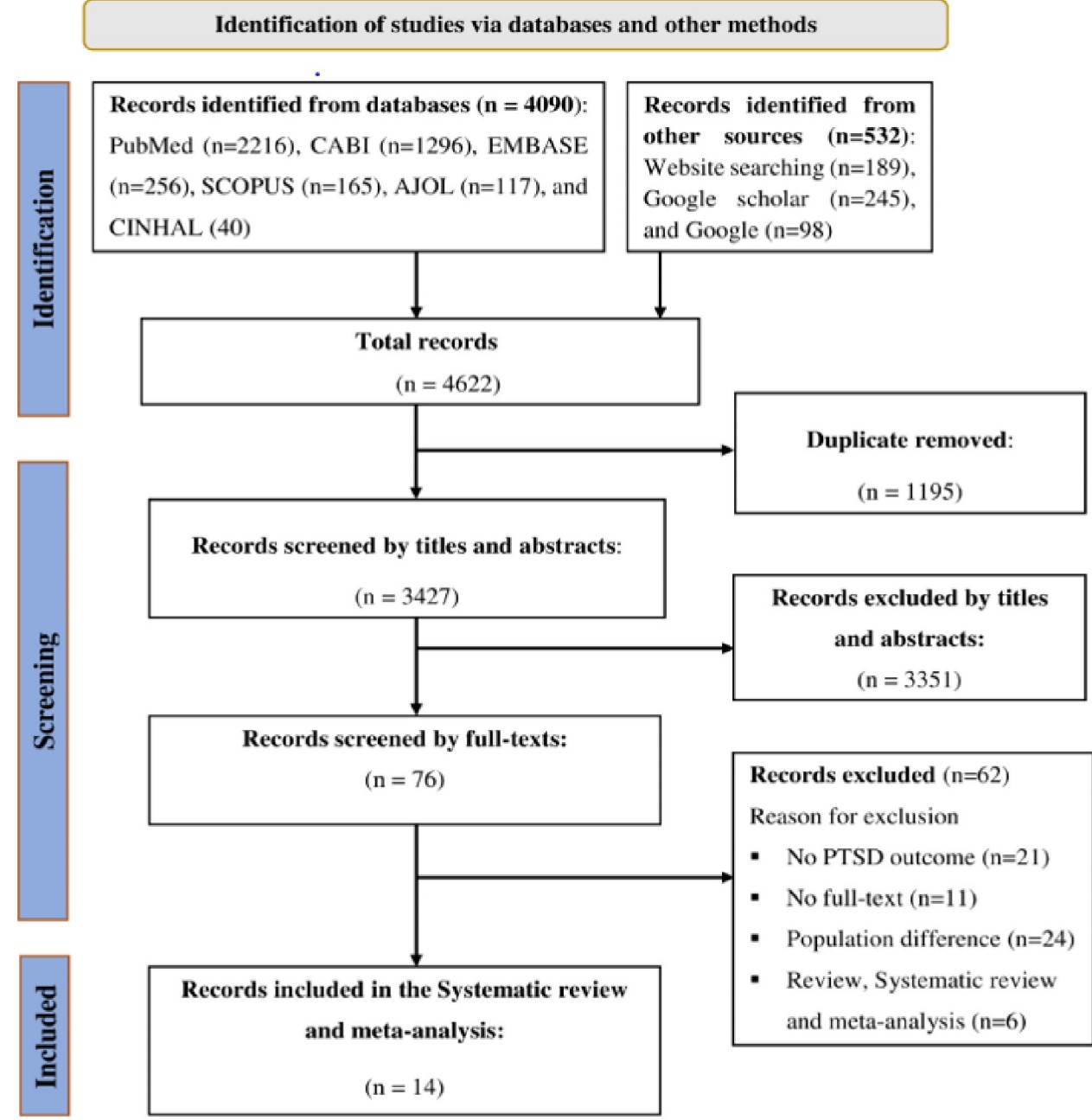

**Fig 1. PRISMA flow diagram for the systematic review and meta-analysis of PTSD and associated factors among IDPs in Africa.**

### Characteristics of the included studies

In this review, the publication year, country of study, study design, sample size, and prevalence of PTSD are summarized in Table 1. By design, all included studies were cross-sectional. This study included a total of 7,590 participants [2,10,20,21,24,26,33–40]. The included articles were published between 2008–2023. The included study sample sizes ranged from 93 to 1291. In this review, a study conducted in Sudan, South Darfur, at the Darfur Campaign study site, showed the lowest prevalence of PTSD (14.9%) [35], while a study conducted in IDP camps in

**Table 1. A description summary of the included articles for the systematic review and meta-analysis on the prevalence and associated factors of PTSD among IDPs in Africa, 2023.**

| Author | Publication year | Country | Study design | Sample size | Response rate | Prevalence (%) | Quality status |
|---|---|---|---|---|---|---|---|
| Asnakew et al [33] | 2019 | Ethiopia | CS | 846 | 98.1 | 37.3 | Low risk |
| Madoro et al [20] | 2020 | Ethiopia | CS | 636 | 98.3 | 58.4 | Low risk |
| Makango et al [26] | 2023 | Ethiopia | CS | 410 | 99.0 | 67.5 | Low risk |
| Masau et al [34] | 2018 | Kenya | CS | 139 | 100.0 | 61.2 | Low risk |
| Ali et al [10] | 2023 | Somalia | CS | 422 | 95.0 | 32.2 | Low risk |
| Elhabiby et al [35] | 2014 | Sudan | CS | 90 | 82.2 | 14.9 | Low risk |
| Roberts et al [36] | 2009 | South Sudan | CS | 1291 | 96.2 | 36.2 | Low risk |
| Roberts et al [21] | 2008 | Uganda | CS | 1280 | 94.5 | 54.3 | Low risk |
| Veling et al [37] | 2013 | DRC | CS | 93 | 100.0 | 40 | Low risk |
| Sheikh et al [24] | 2014 | Nigeria | CS | 228 | 100.0 | 42.2 | Low risk |
| Aluh et al [38] | 2019 | Nigeria | CS | 1200 | 100.0 | 78 | Low risk |
| Ibrahim et al [2] | 2023 | Nigeria | CS | 450 | 100.0 | 94.2 | Low risk |
| Faronbi et al [39] | 2021 | Nigeria | CS | 240 | 100.0 | 56.7 | Low risk |
| Nwoga et al [40] | 2019 | Nigeria | CS | 450 | 93.8 | 37.7 | Low risk |

**Note:** CS: Cross-sectional, DRC: Democratic Republic of Congo.

Yobe State in northeastern Nigeria showed the highest prevalence of PTSD (94.2%) [2]. Three studies were from Ethiopia [20,26,33]; five studies were from Nigeria [2,24,38–40] and the remaining studies were from Kenya [34], Somalia [10], Sudan [35], South Sudan [36], Uganda [21] and the Democratic Republic of Congo (DRC) [37]. The included studies were categorized as having a low risk of bias (quality score 6 to 8). The description of the included studies is presented in Table 1.

**Assessment methods of PTSD.** This research review included studies with varying screening methods for PTSD. While standardized questionnaires were common, most lacked clinical confirmation, raising potential concerns about the accuracy of PTSD diagnoses. However, some studies employed both questionnaires and clinical confirmation, offering a more robust approach to assessing PTSD. Details of these assessment methods used in the original studies are summarized in Table 2.

## Meta-analysis

### Pooled prevalence of PTSD among IDPs in Africa

The pooled prevalence estimate of PTSD was found to be 51% (95% CI: 38–64; $I^2$ = 99.38%). In this analysis, the lowest prevalence of PTSD was found in Sudan at 15% (95% CI: 9–25) [35] and the highest prevalence of PTSD was found in Nigeria at 94% (95% CI: 92–96) [2]. A forest plot shows the prevalence estimates of PTSD among IDPs in Africa (Fig 2).

### Subgroup analysis

Subgroup analysis was done to see the pooled prevalence of PTSD by country. According to the result, of in-country subgroup analysis, the pooled prevalence of PTSD was 62% (95% CI: 41–82) in Nigeria and 54% (95% CI: 36–72) in Ethiopia. Subgroup analysis of the study showed that the highest and lowest prevalence of PTSD was in Nigeria, 62% (95% CI: 41–82), and Sudan, 15% (95% CI: 9–25), respectively (Fig 3). A subgroup analysis was also performed with clinically confirmed cases of PTSD and positive screening cases of PTSD as different

**Table 2. Sampling technique, data collection tool and data collection methods used in the original studies to assess PTSD.**

| Authors, publication year | Sampling technique used to select study participants | Data collection tool used to assess PTSD | Methods of data collection |
|---|---|---|---|
| [b]Asnakew et al, 2019 [33] | Multistage sampling | PCL-C | IA |
| [b]Madoro et al, 2020 [20] | Simple random sampling | PCL-5 | IA |
| [b]Makango et al, 2023 [26] | Systematic random sampling | PCL-5 | IA |
| [b]Masau et al, 2018 [34] | Purposive sampling | NSESSS-PTSD | SA |
| [b]Ali et al, 2023 [10] | Multistage sampling | HTQ | IA |
| [a]Elhabiby et al, 2014 [35] | Purposive sampling | SCID-5-CV | SCI |
| [b]Roberts et al, 2009 [36] | Multistage sampling | HTQ | IA |
| [b]Roberts et al, 2008 [21] | Multistage sampling | HTQ | IA |
| [a]Veling et al, 2013 [37] | Purposive sampling | CIDI-PTSD | SCI |
| [b]Sheikh et al, 2014 [24] | Systematic random sampling | HTQ | IA |
| [b]Aluh et al, 2019 [38] | Purposive sampling | IES-6 questionnaire | IA |
| [b]Ibrahim et al, 2023 [2] | Multistage sampling | HTQ | IA |
| [b]Faronbi et al, 2021 [39] | Multistage sampling | PCL-C | IA |
| [ba]Nwoga et al, 2019 [40] | Systematic random sampling | HTQ and CIDI-PTSD | IA and SCI |

**Note**: **PCL-C** = Post-Traumatic Stress Disorder Checklist for Civilians, **PCL-5** = Post-Traumatic Stress Disorder Checklist for DSM-5, **DSM** = Diagnostic and Statistical Manual of Mental Disorder, **NSESSS-PTSD** = National Stressful Events Survey PTSD Short Scale, **HTQ** = Harvard Trauma Questionnaire, **SCID-5-CV** = Structured Clinical Interview for DSM-5 Disorders: Clinician Version, **CIDI** = Composite International Diagnostic Interview, **IES** = Impact of Event Scale-6, **IA** = Interviewer-administered, **SA** = Self-administered, **SCI** = Structured clinical interview.

[a] = Studies assess PTSD with clinical confirmation.

[b] = Studies assess PTSD without clinical confirmation.

[ba] = Studies assess PTSD using a mix of both assessments.

subgroups. Accordingly, the pooled prevalence of clinically confirmed cases of PTSD was 31% (95% CI: 15–46) and positive screening cases of PTSD was 55% (95% CI: 44–65) (Figs 4 & 5).

## Heterogeneity and publication bias

The presence of heterogeneity and publication bias (small study effect) was assessed within the included studies. The included studies had a high degree of heterogeneity ($I^2$ = 99.38%, p = 0.00). Publication bias was assessed using a funnel plot and Egger's regression test at a p-value <0.05. The funnel plot showed that the distribution of studies was asymmetrical, whereas Egger's test was found to be not statistically significant for the estimated prevalence of PTSD (p = 0.063), meaning that there was no evidence of publication bias (Fig 6).

## Sensitivity analysis test

A sensitivity analysis was performed to assess the effect of each study on the pooled estimate of PTSD. However, the results of the sensitivity analysis showed that there was no single study effect on the pooled prevalence of PTSD in the fitted meta-analytic model, as shown in Fig 7.

**Factors associated with PTSD among IDPs in Africa.** Factors associated with PTSD were identified based on the pooled effect of two or more studies. In this meta-analysis, factors associated with PTSD were assessed using 14 studies [2,10,20,21,24,26,33–40]. The analysis showed that in 4 of these studies [20,21,33,36], female IDPs were found to have a twofold higher risk of developing PTSD compared to male IDPs (OR = 1.99, 95% CI: 1.65–2.32). The pooled results of three studies [21,33,36] revealed that individuals who were no longer married (divorced, separated, widowed, or forcefully separated) had 1.93 times higher likelihood of

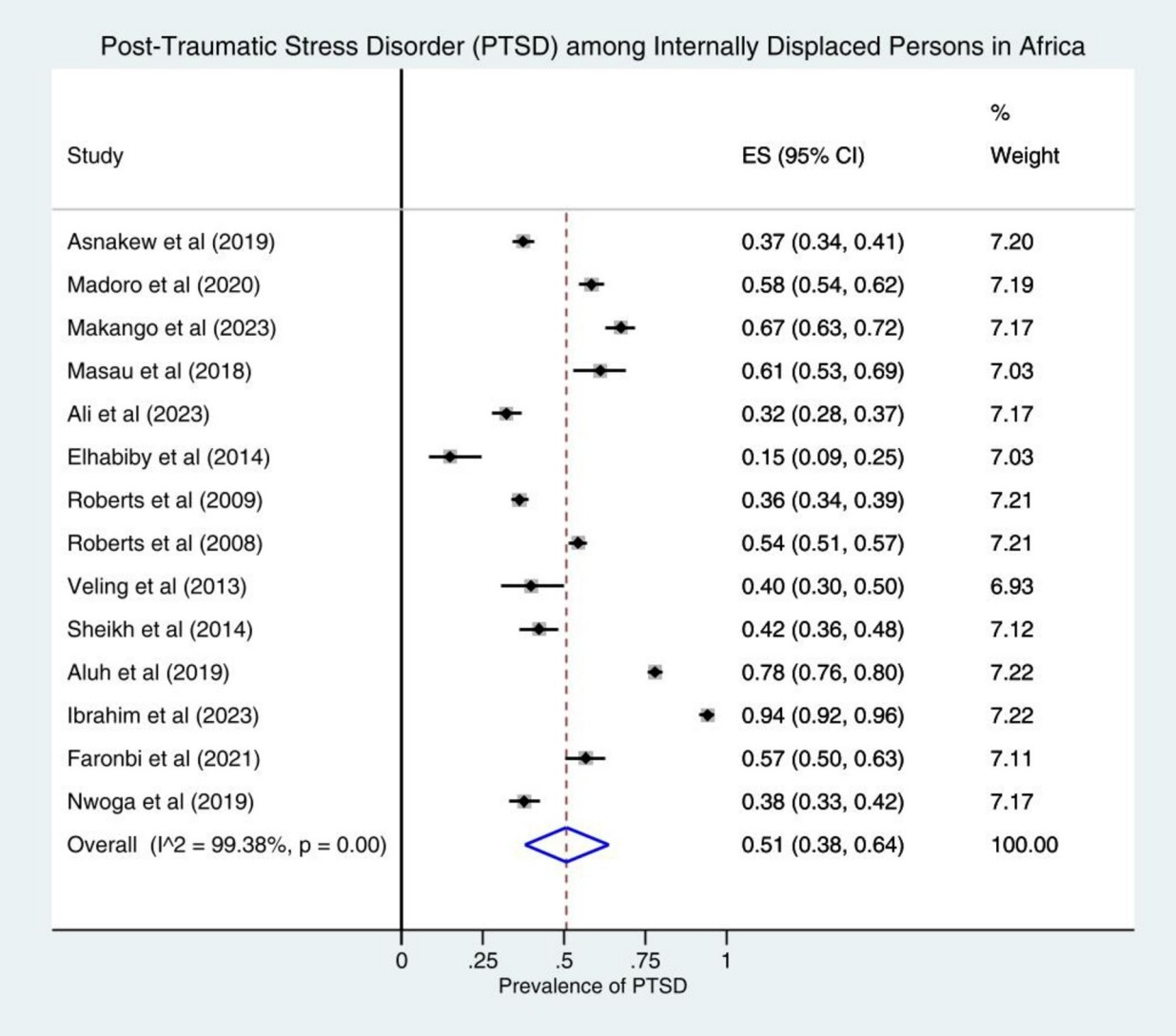

**Fig 2. Forest plot of prevalence of PTSD among IDPs in Africa, 2023.**

PTSD compared to those who were married or single (OR = 1.93, 95% CI: 1.43–2.43). Similarly, the pooled findings of two studies [10,26] showed that the likelihood of PTSD was 1.92 times higher for unemployed IDPs compared to employed IDPs (OR = 1.92, 95% CI: 1.17–2.67). Furthermore, two studies' combined results [33,36] revealed that the likelihood of PTSD was 1.94 times higher for injured IDPs than for uninjured ones (OR = 1.94, 95% CI: 1.50–2.37).

Moreover, the pooled results of two studies [20,21] of this meta-analysis revealed a positive correlation between the likelihood of developing PTSD and the commutative number of traumatic incidents encountered. The odds of PTSD were higher in IDPs who had experienced four or more of the sixteen traumatic events (OR = 2.09, 95% CI: 1.16–3.01), 3.15 times higher in those who had experienced eight to eleven traumatic events (OR = 2.09, 95% CI: 2.18–4.12), and 5.37 times higher in those who had experienced twelve or more traumatic events (OR = 5.37, 95% CI: 2.61–8.12) compared to those who had experienced zero to three

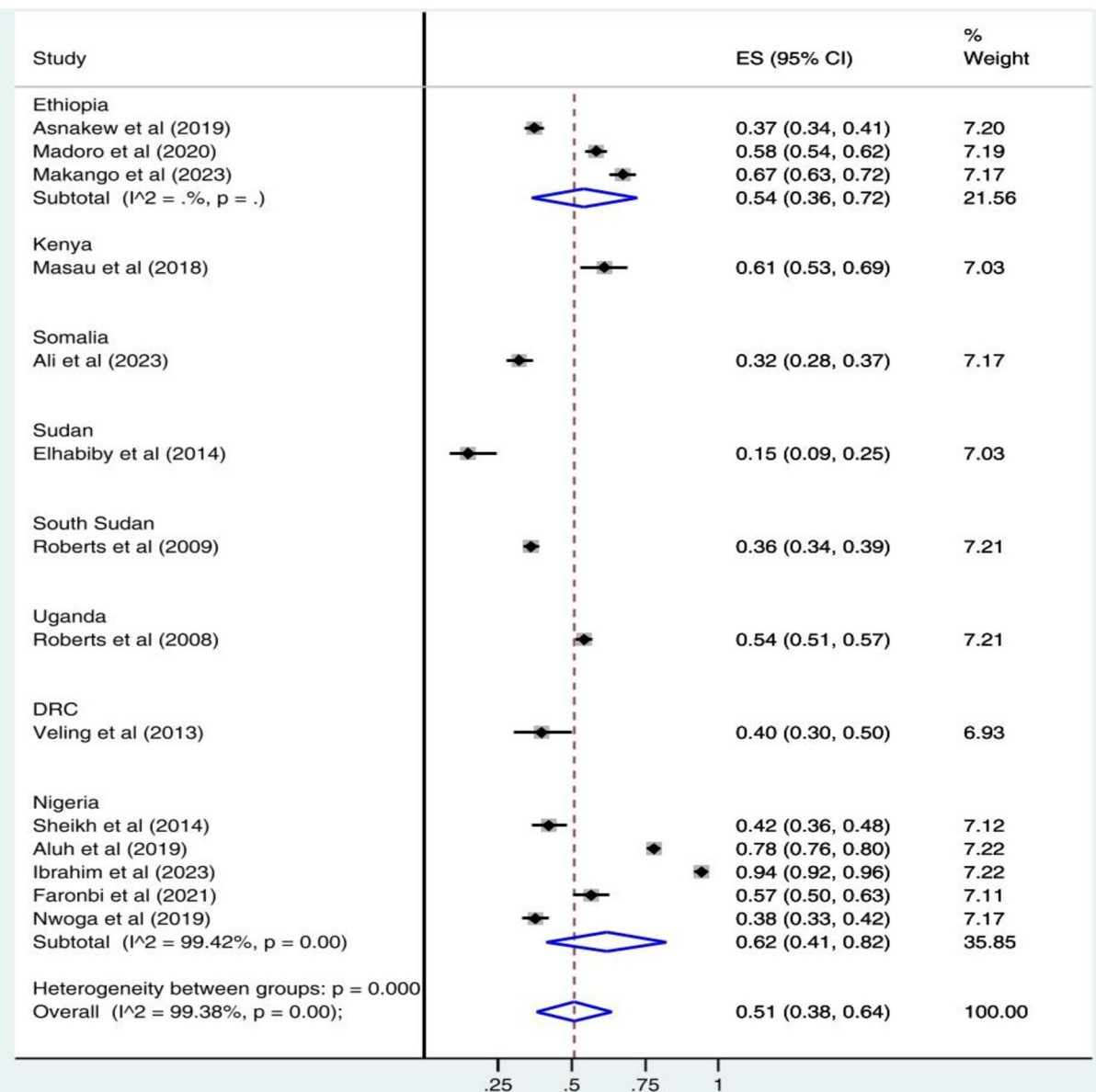

**Fig 3. Subgroup analysis by country for the pooled proportion of PTSD among IDPs in Africa.**

traumatic events. The combined findings of two studies [21,36] revealed that the odds of having PTSD were 1.92 times higher for IDPs with poor health who did not receive medical care than for those who did receive medical care (OR = 1.92, 95% CI: 1.41–2.29). Furthermore, the pooled result from four studies [2,20,24,26] revealed that people with depression had a 2.97-fold increased risk of developing PTSD compared to people without depression (OR = 2.97, 95% CI: 2.07–3.86). Additionally, the current analysis discovered a substantial correlation between PTSD and a higher frequency of displacement. The meta-analysis's combined findings showed that those who were internally relocated more than once had a 2.13-fold increased risk of developing PTSD (Table 3).

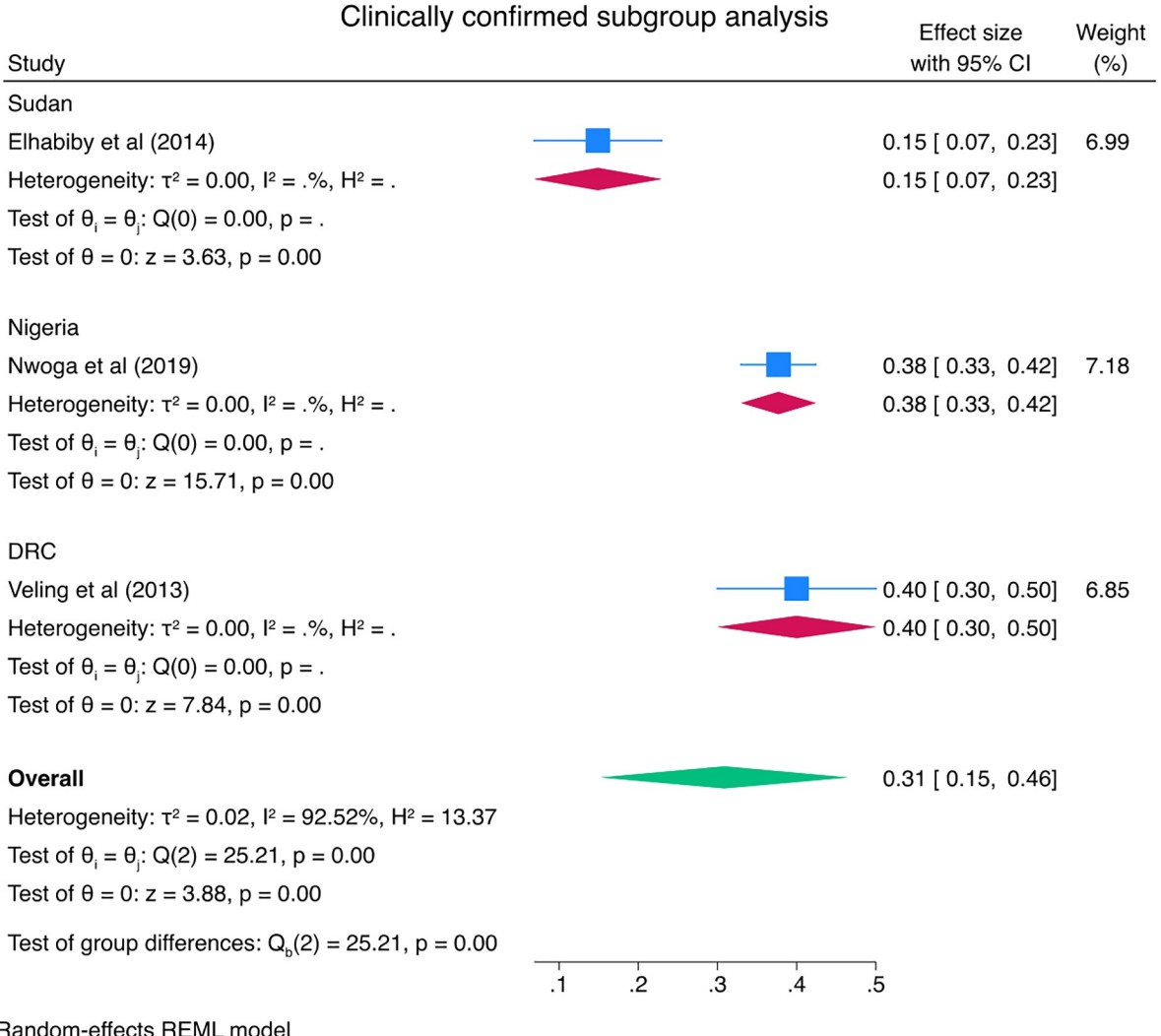

**Fig 4. Subgroup analysis by clinically confirmed cases of PTSD among IDPs in Africa.**

## Discussion

Internally displaced persons (IDPs) are particularly vulnerable to PTSD because they have often experienced traumatic events such as violence, loss of loved ones, and destruction of their homes [41,42]. They may also have difficulty accessing mental health care, which can make it harder to recover from PTSD. Investigating the overall impact and risk factors might be important to the development of interventions to prevent and treat PTSD among IDPs in Africa. The current review presents comprehensive findings on the pooled magnitude of PTSD and its associated factors among IDPs in Africa.

This systematic review and meta-analysis found that the pooled prevalence of PTSD among IDPs in Africa was 51% (95% CI: 38–64%). The prevalence of PTSD in this review was aligned with the studies carried out in the Kurdistan region of Iraq (60%) [43], and Sri Lanka (56%) [44]. Moreover, this finding is in concordance with a systematic review and meta-analysis

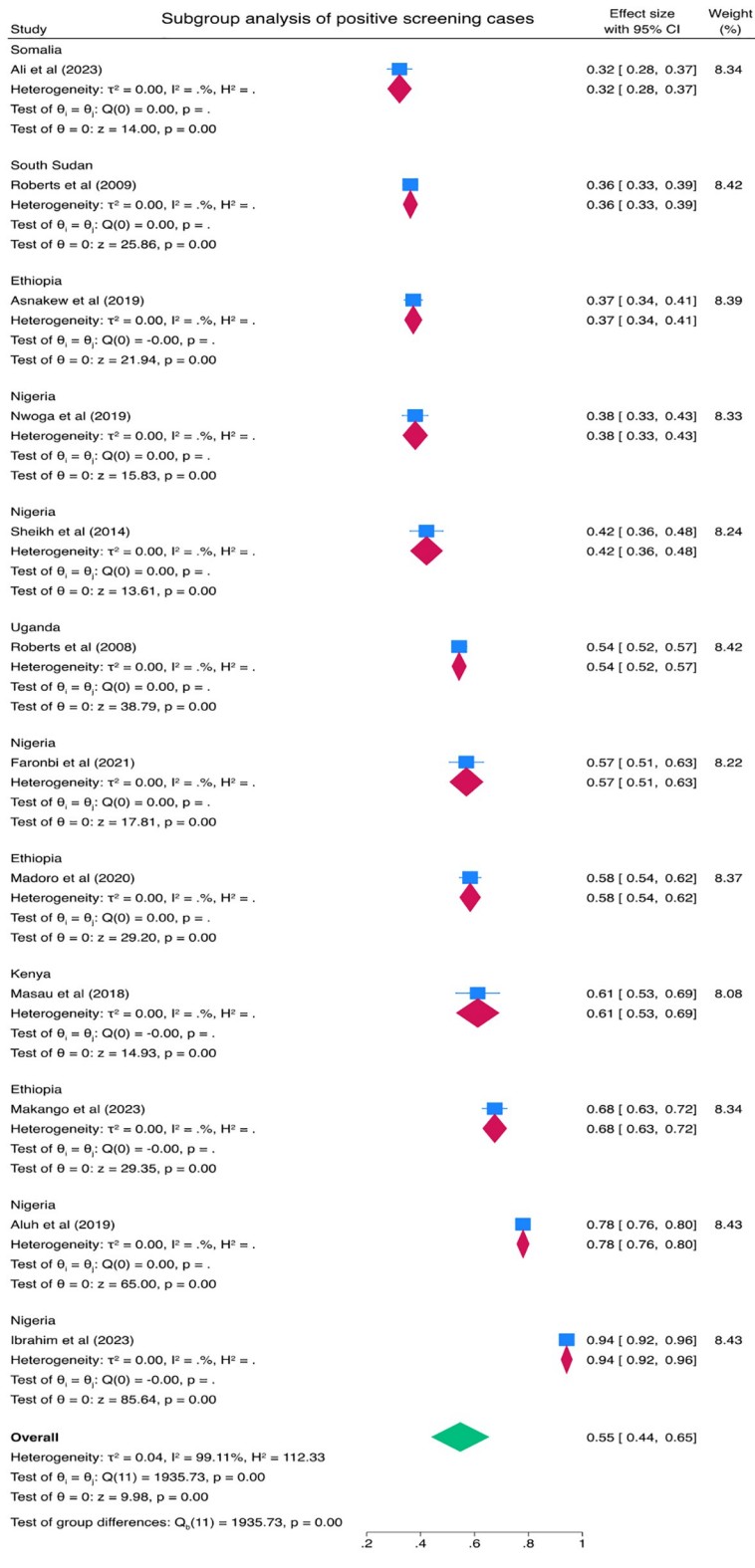

**Fig 5. Subgroup analysis by positive screening cases of PTSD among IDPs in Africa.**

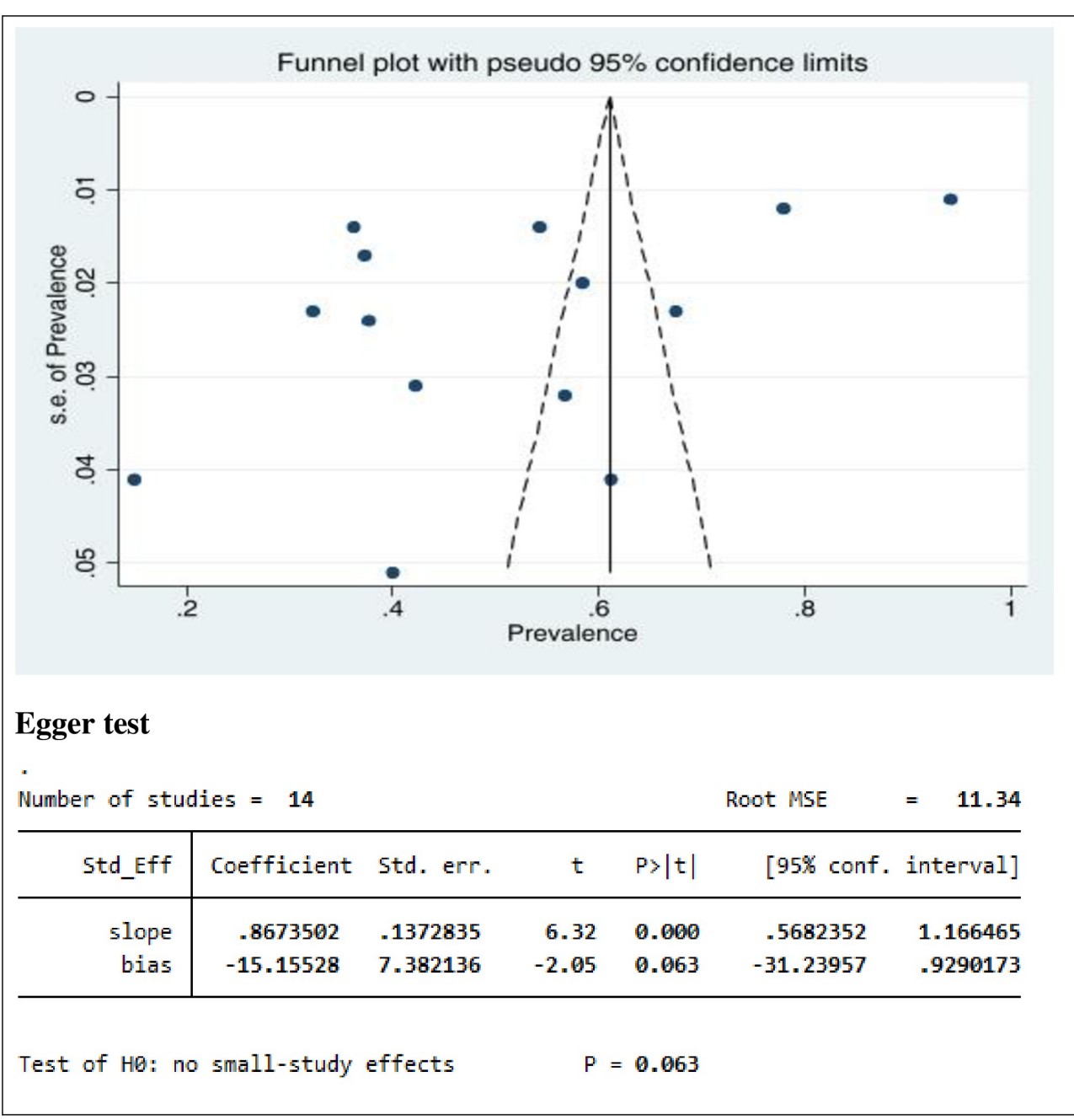

**Fig 6. Funnel plot and Egger's test of studies reporting PTSD among IDPs in Africa.**

study in Syria which reported a pooled estimate of 36% PTSD [45]. On the other hand, the prevalence of PTSD in this review was lower than that of the study done in Medellin, Colombia (88%) [46]. There are several possible explanations for these disparities. One possibility could be a difference in methodological approaches. Another possibility is that the studies were conducted with different populations in different cultural and social contexts to manage problems related to displacements [47]. Antithetically, the estimated prevalence of PTSD in the current review was higher than the studies carried out in another study in Sri Lanka (2.3%) [48],

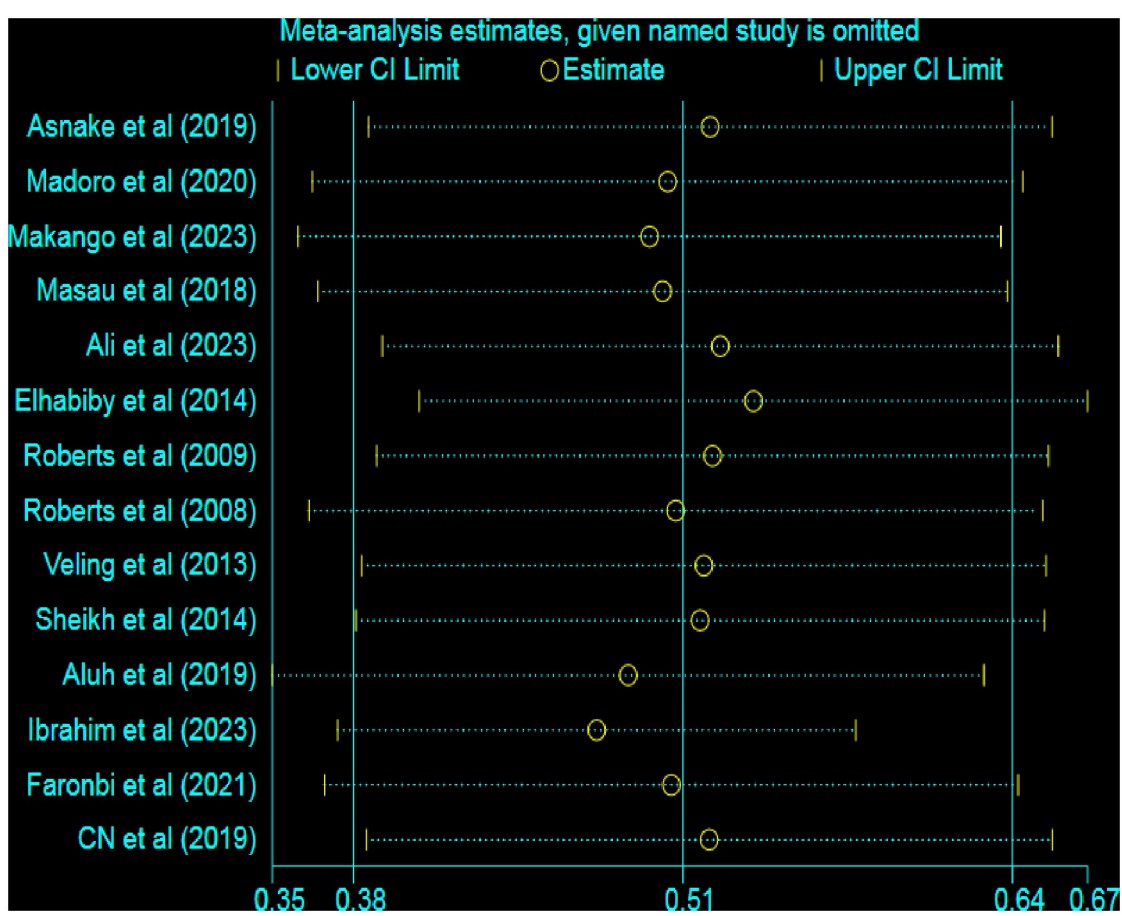

**Fig 7. Sensitivity analysis graph to examine the effect of each study on the pooled result.**

Georgia (23.3%) [49], Iraq (20.8%) [50], and India (9%) [51]. Furthermore, the prevalence of PTSD in the current study is higher than in a systematic review and meta-analysis of an epidemiological study done by Nexhmedin M. et al which reported a 26% pooled prevalence of PTSD among survivors living in war-afflicted regions [52].

The subgroup meta-analysis of this review showed that the pooled prevalence of clinically confirmed cases of PTSD was 31% (95% CI: 15–46) and positive screening cases of PTSD was 55% (95% CI: 44–65). One possible explanation for the higher prevalence rate among positive screening cases (55%) compared to clinically confirmed cases (31%) could be related to the sensitivity of the screening tool used. Screening tools are designed to identify individuals who may be at risk or likely to have the condition, but they may also capture individuals with false positives those who screen positive but may not meet the diagnostic criteria upon further clinical assessment. These false positives could inflate the prevalence rate among positive screening cases. On the other hand, clinically confirmed cases undergo a more comprehensive diagnostic evaluation, which includes clinical interviews, symptom assessment, and adherence to specific diagnostic criteria. This process is typically conducted by trained professionals such as qualified mental health professional and aims to ensure a more accurate diagnosis of PTSD. By applying stricter criteria, the diagnostic process may exclude individuals who had initially screened positive but do not meet the clinical diagnostic threshold. Consequently, the

Table 3. Factors associated with PTSD among IDPs in Africa, 2023.

| Factors (Reference) | Authors (year) and $I^2$ with p-value | Odds ratio with 95% CI | % Weight |
|---|---|---|---|
| Sex (Male) | Asnakew et al (2019) [33] | 1.70 (1.20, 2.50) | 26.79 |
| | Madoro et al (2020) [20] | 2.35 (1.61, 3.44) | 13.52 |
| | Roberts et al (2009) [36] | 2.01 (1.52, 2.66) | 34.84 |
| | Roberts et al (2008) [21] | 2.06 (1.49, 2.84) | 24.84 |
| | Overall, IV ($I^2$ = 0.0%, p = 0.704) | **1.99 (1.65, 2.32)** | 100.00 |
| No longer married* | Asnakew et al (2019) [33] | 2.10 (1.30, 3.40) | 22.72 |
| | Roberts et al (2009) [36] | 2.10 (1.28, 3.44) | 21.48 |
| | Roberts et al (2008) [21] | 1.80 (1.25, 2.59) | 55.80 |
| | Overall, IV ($I^2$ = 0.0%, p = 0.843) | **1.93 (1.43, 2.43)** | 100.00 |
| Employment status (Employed) | Makango et al (2021) [26] | 2.09 (1.24, 3.54) | 42.56 |
| | Ali et al (2021) [10] | 1.79 (1.06, 3.04) | 57.44 |
| | Overall, IV ($I^2$ = 0.0%, p = 0.698) | **1.92 (1.17, 2.67)** | 100.00 |
| Being injured (No) | Asnakew et al (2019) [33] | 8.30 (5.00, 13.60) | 1.01 |
| | Roberts et al (2009) [36] | 1.87 (1.49, 2.36) | 98.99 |
| | Overall, IV ($I^2$ = 88.2%, p = 0.004) | **1.94 (1.50, 2.37)** | 100.00 |
| Number of cumulative trauma events (0–3 trauma) | | | |
| 4–7 trauma events | Madoro et al (2020) [20] | 1.90 (1.10, 3.40) | 64.50 |
| | Roberts et al (2008) [21] | 2.43 (1.30, 4.40) | 35.50 |
| | Overall, IV ($I^2$ = 0.0%, p = 0.590) | **2.09 (1.16, 3.01)** | 100.00 |
| 8–11 trauma events | Madoro et al (2020) [20] | 2.90 (2.00, 4.10) | 85.50 |
| | Roberts et al (2008) [21] | 4.62 (2.70, 7.80) | 14.50 |
| | Overall, IV ($I^2$ = 33.1%, p = 0.222) | **3.15 (2.18, 4.12)** | 100.00 |
| 12–16 trauma events | Madoro et al (2020) [20] | 4.10 (1.70, 9.70) | 47.44 |
| | Roberts et al (2008) [21] | 6.51 (3.70, 11.30) | 52.56 |
| | Overall, IV ($I^2$ = 0.0%, p = 0.392) | **5.37 (2.61, 8.12)** | 100.00 |
| Ill health without medical care (No) | Roberts et al (2009) [36] | 1.89 (1.41, 2.53) | 43.86 |
| | Roberts et al (2008) [21] | 1.95 (1.52, 2.51) | 56.14 |
| | Overall, IV ($I^2$ = 0.0%, p = 0.875) | **1.92 (1.55, 2.29)** | 100.00 |
| Depression (No) | Madoro et al (2020) [20] | 2.60 (1.79, 3.78) | 80.80 |
| | Makango et al (2023) [26] | 5.42 (3.20, 8.31) | 9.51 |
| | Sheikh et al (2014) [24] | 3.53 (1.66, 7.48) | 9.47 |
| | Ibrahim et al (2023) [2] | 7.00 (1.30, 37.90) | 0.22 |
| | Overall, IV ($I^2$ = 21.7%, p = 0.280) | **2.97 (2.07, 3.86)** | 100.00 |
| Frequency of the displacement (Once) | Madoro et al (2020) [20] | 3.69 (2.35, 5.82) | 17.17 |
| | Roberts et al (2009) [36] | 1.81 (1.18, 2.76) | 82.83 |
| | Overall, IV ($I^2$ = 73.2%, p = 0.053) | **2.13 (1.41, 2.85)** | 100.00 |

Key:

* No longer married (divorced/separated, widowed, forcefully separated) compared to married and never married.

prevalence rate among clinically confirmed cases may be lower compared to positive screening cases [53–55].

Several factors may also contribute to the differences. The first possible reason could be due to the impact of conflict, mass displacement, and the level of property destruction repeatedly encountered in the African region where this review was conducted. This may include a combination of war and political and ethnic conflict [22]. In addition, this difference may be

related to the differences in the accessibility and affordability of mental health care services in those different settings [23]. Furthermore, it is also possible that the studies were conducted at different times, and that the prevalence of PTSD has changed over time.

A meta-analysis of this review found that females were at higher risk of PTSD than men. Several factors could contribute to the higher prevalence of PTSD among females compared to males. Numerous studies across different countries have documented the higher prevalence of PTSD among females compared to males [13,23,56–60]. These factors can broadly be related to biological, psychological, and social factors. Hormonal differences between females and males that estrogen may enhance emotional reactivity and memory consolidation, potentially increasing the risk of PTSD [61]. Social factors could also contribute to increased PTSD in females because females are more likely to experience interpersonal trauma, such as sexual assault or domestic violence compared with males [23,62,63], which are associated with a higher risk of PTSD compared to other types of traumas. Gender roles and expectations of females may also emphasize emotional expressiveness and vulnerability, which could make them more susceptible to developing PTSD symptoms. Furthermore, psychological factors like coping strategies and social support might be among the factors contributing to the occurrence of PTSD among female IDPs. Females may be more likely to use emotion-focused coping strategies, such as rumination and suppression, which may exacerbate PTSD symptoms. Males, on the other hand, may favor problem-focused coping, which may be more effective at managing stress. In addition, females often have stronger social networks, which can provide emotional support and buffer against stress. However, these networks may also expose females to more trauma-related discussions, potentially increasing the risk of PTSD. Therefore, understanding these factors is crucial for developing effective prevention and treatment strategies for PTSD in this population. This finding may also suggest that it could be important to give more attention to females to provide psychological and general support to minimize the impact on their daily lives.

In this review, being divorced, separated, and widowed was found to be more associated with PTSD compared with being married and/or single. The finding is also consistent with previous studies [23,64]. Marital status can influence an individual's vulnerability to PTSD in several ways: social support and emotional buffer, joint problem-solving and resource sharing, and sociocultural factors that marriage can provide a strong source of social support and emotional buffering, which can help individuals cope with stress and trauma. These findings suggest that marital status can play a significant role in influencing the risk of PTSD among IDPs. The supportive and protective aspects of marriage can help individuals cope with the challenges of displacement and trauma, reducing their vulnerability to PTSD. However, it is important to note that the relationship between marital status and PTSD may not always be straightforward. In some cases, some studies showed that marital strain or conflict can increase the risk of PTSD among married individuals [65,66]. Therefore, the impact of marital status on PTSD may vary depending on individual circumstances, cultural factors, and the specific nature of the traumatic event.

Consistent with earlier findings [10], employment status was also found to have a significant association with PTSD among IDPs. Several factors contribute to this association: loss of routine and structure that unemployment disrupts an individual's daily routine and structure, which can be particularly destabilizing for IDPs who have already experienced the upheaval of displacement; financial strain, and economic hardship, adding to the stress and anxiety of displacement; limited access to mental health care that can hinder an individual's ability to access mental health care, either due to financial constraints or the lack of employer-sponsored health insurance. These findings underscore the importance of employment in promoting mental health and well-being among IDPs. Supporting employment opportunities and providing

vocational training can help IDPs regain a sense of normalcy, purpose, and control, reducing their risk of developing PTSD and improving their overall quality of life.

This review revealed that being injured is significantly associated with the existence of PTSD among IDPs, which is in line with other studies [23,42,67]. Experiencing physical injury during displacement can significantly increase an individual's risk of developing PTSD because of direct trauma and physical pain, psychological impact of injury, impact on daily living and functioning, limited access to healthcare and support, trauma reactivation, and reminders of injury. These findings highlight the importance of addressing physical injuries and providing comprehensive support services for IDPs to mitigate the risk of developing PTSD. Timely and effective medical care, rehabilitation services, and psychological support can significantly improve the physical and mental health outcomes of IDPs who have experienced injuries.

The current finding has shown, consistent with earlier evidence [23,45,68,69], that the number of trauma events experienced was positively associated with the risk of PTSD. There is a well-established relationship between the number of trauma events experienced and the risk of PTSD among IDPs. Research suggests that each additional trauma event increases the likelihood of developing PTSD. This association can be attributed to several factors, such as accumulation of stress and emotional overload. Each trauma event exposes an individual to significant stress and emotional overload, taxing their coping mechanisms and increasing their vulnerability to developing PTSD. The cumulative effect of multiple trauma events can overwhelm an individual's ability to process and integrate these experiences, leading to the development of PTSD symptoms. In addition, repeated exposure to traumatic events can heighten an individual's sensitivity to trauma cues, making them more likely to experience flashbacks, nightmares, and intrusive thoughts related to the traumatic experiences. This sensitization can maintain the state of hypervigilance and emotional arousal characteristic of PTSD. These findings may underscore the importance of preventing and addressing trauma exposure among IDPs to reduce the risk of developing PTSD. Providing early intervention and mental health support services can help IDPs cope with the effects of trauma and prevent the development of PTSD.

Illness in the absence of medical care was found to be significantly associated with the presence of PTSD among IDPs and this is in line with other findings [21]. The absence of medical care significantly elevates the risk of developing PTSD among IDPs. This association stems from several interconnected factors including unmet physical and psychological needs. Lack of access to medical care can lead to the neglect of both physical and psychological needs following a traumatic event. These findings highlight the importance of providing comprehensive healthcare services to IDPs, including both physical and mental healthcare. Addressing their medical needs can significantly reduce the risk of developing PTSD and promote their overall well-being.

In the current review, the presence of depression is found to have a significant association with PTSD among IDPs, which is in line with earlier studies [10,21,23]. The co-occurrence of depression and post-traumatic stress disorder (PTSD) is prevalent among internally displaced persons (IDPs). This could be because of shared etiological factors that both depression and PTSD share common risk factors, such as exposure to trauma, genetic predisposition, and neurobiological alterations. Both depression and PTSD involve dysregulation of stress hormones, such as cortisol and norepinephrine. These hormonal imbalances can contribute to the development and maintenance of both conditions. In addition, IDPs face a range of social and environmental stressors, such as displacement, loss of social support, and economic hardship. These stressors can contribute to both depression and PTSD by exacerbating emotional distress and reducing resilience. These findings implicate the importance of addressing both

depression and PTSD simultaneously in IDPs. Integrated treatment approaches that target both conditions can significantly improve the mental health outcomes of IDPs.

The current review also revealed that an increased frequency of displacement was also found to be significantly associated with PTSD. The finding is consistent with previous studies [21,23,42,67,68]. The number of times an individual had been displaced was positively associated with the risk of PTSD. Multiple factors contribute to the increased risk of PTSD among IDPs who experience repeated displacements. This could be because each displacement experience exposes an individual to significant stress and emotional overload, taxing their coping mechanisms and increasing their vulnerability to developing PTSD. Repeated displacements lead to an accumulation of traumatic experiences, making it increasingly difficult to process and integrate these experiences, ultimately increasing the risk of PTSD. Frequent displacements can heighten an individual's sensitivity to trauma cues, making them more likely to experience flashbacks, nightmares, and intrusive thoughts related to the traumatic experiences. This sensitization perpetuates the state of hypervigilance and emotional arousal characteristic of PTSD. These findings suggest the importance of preventing and addressing repeated displacements to reduce the risk of PTSD among IDPs. Providing stable housing, livelihood support, and mental health services can help IDPs cope with the effects of displacement and prevent the development of PTSD.

## Strengths and limitations of the study

This study was a first-of-its-kind systematic review and meta-analysis that estimated the pooled prevalence and associated risk factors of PTSD among IDPs in Africa. The study identified several significant risk factors for PTSD among IDPs. This information can be used to develop targeted interventions to prevent PTSD. This study has its limitations. Firstly, it was a cross-sectional study, so it could not establish cause-effect relationships. In addition, the lack of studies from countries other than those included may limit the continental representativeness of the study. Overall, the study is a valuable contribution to our understanding of PTSD among IDPs in Africa. The findings can be used to inform the development of targeted interventions to prevent PTSD among this vulnerable population.

## Conclusions

The findings of this systematic review and meta-analysis highlight the alarming prevalence of PTSD among IDPs in Africa. The estimated pooled prevalence of 51% is significantly higher than the general population prevalence of PTSD, demonstrating the unique challenges faced by IDPs in coping with trauma and displacement. The study's identification of significant risk factors, including female gender, marital status, traumatic events, ill health without medical care, depression, and frequency of displacement, provides valuable insights for targeted interventions. Effective interventions and the development of tailored mental health programs are needed to prevent and treat PTSD among IDPs, with focusing on the identified risk factors. Future studies focusing on the determinant factors of PTSD and their impacts on IDPs need to be welcomed.

## Supporting information

**S1 File. PRISMA Checklist used in the reports of systematic review and meta-analysis.**
(DOCX)

**S2 File. JBI quality appraisal/result of the quality assessment of the studies.**
(DOCX)

**S3 File. Data set used in generating and analyzing of systematic review and meta-analysis.** (DTA)

## Acknowledgments

The authors would like to thank the University of Gondar, Ethiopia, for providing an office and free internet service. Moreover, the authors thanked and recognized the articles included in this study and used them as a basis for this systematic review and meta-analysis.

## Author Contributions

**Conceptualization:** Amensisa Hailu Tesfaye.

**Data curation:** Amensisa Hailu Tesfaye, Girum Shibeshi Argaw, Belay Desye, Abiy Ayele Angelo, Fantu Mamo Aragaw, Giziew Abere.

**Formal analysis:** Amensisa Hailu Tesfaye, Ashenafi Kibret Sendekie, Fantu Mamo Aragaw.

**Investigation:** Amensisa Hailu Tesfaye, Ashenafi Kibret Sendekie, Girum Shibeshi Argaw, Fantu Mamo Aragaw, Giziew Abere.

**Methodology:** Amensisa Hailu Tesfaye, Ashenafi Kibret Sendekie, Gebisa Guyasa Kabito, Garedew Tadege Engdaw, Belay Desye, Abiy Ayele Angelo, Fantu Mamo Aragaw, Giziew Abere.

**Software:** Amensisa Hailu Tesfaye.

**Supervision:** Garedew Tadege Engdaw, Belay Desye, Giziew Abere.

**Validation:** Amensisa Hailu Tesfaye, Ashenafi Kibret Sendekie, Abiy Ayele Angelo, Giziew Abere.

**Visualization:** Amensisa Hailu Tesfaye, Ashenafi Kibret Sendekie, Gebisa Guyasa Kabito, Garedew Tadege Engdaw, Girum Shibeshi Argaw, Belay Desye, Abiy Ayele Angelo, Giziew Abere.

**Writing – original draft:** Amensisa Hailu Tesfaye, Ashenafi Kibret Sendekie.

**Writing – review & editing:** Amensisa Hailu Tesfaye, Gebisa Guyasa Kabito, Garedew Tadege Engdaw, Girum Shibeshi Argaw, Belay Desye, Abiy Ayele Angelo, Fantu Mamo Aragaw, Giziew Abere.

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
