## [Decision Letter · Decision Letter 0]

15 Jan 2024

PONE-D-23-40921Post-traumatic stress disorder and associated factors among internally displaced persons in Africa: A systematic review and meta-analysisPLOS ONE

Dear Dr. Tesfaye,

Thank you for submitting your manuscript to PLOS ONE. After careful consideration, we feel that it has merit but does not fully meet PLOS ONE’s publication criteria as it currently stands. Therefore, we invite you to submit a revised version of the manuscript that addresses the points raised during the review process.

Kindly address all reviewers' recommendations.

We look forward to receiving your revised manuscript.

Kind regards,

Roberto Ariel Abeldaño Zuñiga

Academic Editor

PLOS ONE

Journal Requirements:

https://doi.org/10.1371/journal.pone.0287996

In your revision ensure you cite all your sources (including your own works), and quote or rephrase any duplicated text outside the methods section. Further consideration is dependent on these concerns being addressed.

Additional Editor Comments:

Kindly address all reviewers' recommendations.

Reviewers' comments:

Reviewer's Responses to Questions

**Comments to the Author**

1. Is the manuscript technically sound, and do the data support the conclusions?

Reviewer #1: Yes

Reviewer #2: Yes

2. Has the statistical analysis been performed appropriately and rigorously? 

Reviewer #1: Yes

Reviewer #2: Yes

3. Have the authors made all data underlying the findings in their manuscript fully available?

Reviewer #1: Yes

Reviewer #2: Yes

4. Is the manuscript presented in an intelligible fashion and written in standard English?

Reviewer #1: No

Reviewer #2: Yes

5. Review Comments to the Author

Reviewer #1: General

Thank you authors for coming with this interesting topic. I think, The finding from the current study will be very important for the regional policy makers. in order to further improve the manuscript, authors need to consider some comments listed below. Initially, authors are suggested to go through their document and edit it for some spelling error and repetition of sentences comprising the same message/idea. Furthermore, Authors are also advised to use Abbreviation consistently in the document once decided to do so.

Abstract

Method: sentence number 1 and 2 have similar message/idea, so authors are advised to merge them in one sentence.

Result: 4-7, 8-11 and 12-16 traumatic events were significantly associated with PTSD. However it is not clear weather this number indicates number of traumatic events experienced or frequency of exposure to traumatic events, hence, the two were conceptually different. e.g an individual may be exposed to various traumatic events at single occasion or may experience repeated traumatic events on different occasions. Moreover, the three category (4-7,8-11,12-16) were reported to be associated with PTSD, unless you are interested in examining which category is highly significant, you can merge them as ≥ 4 traumatic events.

Introduction

Line 63 “PTSD is a serious” the word serious is a vague word, please be advised to replace it with other appropriate word.

Line 74 “insufficient mental health professional” better replaced with “insufficient mental health services” because the latter one can best address both facility and professionals.

Line 92-95: too long sentence and difficult to grasp its meaning. Please rewrite it.

Line 102-103: “a comprehensive study that can address the overall public health impact of IDPs in terms of causing PTSD…” please replace IDPs with internal displacement, otherwise it gives meaning other than the intended one.

Method

Line 122-123: the sentence begin with “the following database”… lacks to give full meaning please revise it.

Line 123-125: please delete “the search strategy aimed to identify studies reporting on PTSD prevalence rates and associated factors among internally displaced persons (IDPs) in Africa”.

Inclusion criteria: I suggest you to merge and write it as paragraph, not listed bullet

Line 148: please delete “in this study research articles like”

Line 176-177: “it was evaluated using the JBI” repetition of idea in prior sentence so delete it.

Line 183: please delete “there are two main outcomes for this review” because the readers can easily understand this from the title

191: The I2 score range used to categorize the degree of heterogeneity overlap each other, and I think this is not correct. Authors are suggested to revise it. May be you can refer to https://doi.org/10.1007/s13312-022-2500-y

Result

Please revise your result for consistency, logical flow of ideas and typos. There is unnecessary detail throughout the result section particularly under factors associated with PTSD. There is detailed information both in text narration and tables, thus I suggest authors to minimize duplication of information.

Result presented in the same style e.g. “unemployed were 1.92 times more likely to have PTSD than employed IDPs... “, this style appeared to be commonly used throughout the document. In order to make the document more attractive and help the readers to make sense of it, it is crucial to present the result in different, clear and intelligent fashion. I recommend authors to go through the document and make some amendments with regard to this point.

On quality appraisal of the included studies, I think the authors need to clearly describe full result (each assessment question with corresponding given score) to ensure that detailed objective scoring are disclosed beforehand and will be reproduced.

Line 213-214: “the included articles were conducted b/n 2008-2023” please replace “conducted” with published. Because you don’t know when the studies was carried out but when the article is published.

Line 220 - Please delete “were used to determine the pooled prevalence of PTSD among IDPs.”

Line 228-229 - the sentence has to be stated in method section.

Line 259 there is unnecessary detail, I suggest the authors to directly state finding from the included studies, please delete “Based on the results of the 14 included studies, the analysis showed that”

Line 261 please revise punctuation error

Line 263: please delete “being exposed to”, and elsewhere in the document please replace more likely to “develop” with appropriate word, because with pooled prevalence result from cross sectional studies it is difficult to talk about incidence (“to develop” is in favor of new incidence)

Line 269 and elsewhere in documents please carefully revise spelling error eg. Metal analysis to meta-analysis.

Discussion

Authors have tried to explain the possible justification toward the discrepancy between this study and other study finding. However it is not clear whether the stated justification is to higher or lower prevalence rate, also it is not clearly stated on how those factors may account to the observed difference. Moreover, only one reference is cited to indicate evidence that supports their long justification, which is inappropriate as to me.

Line 317-321 almost repetition of result, please delete it

Line 344: “exposed to” better replaced by ‘associated with’

Line 358-359 please delete it because it is repetition “Employment status can significantly impact an individual's risk of developing PTSD among IDPs”.

Strength and limitation

Line 439: “The study's strengths outweigh its limitations”. I suggest authors to revise/delete this sentence, as to me it is not reasonable to claim this at this time.

Reviewer #2: This is an interesting paper and the content is very good but I suggest the paper can be improved in the following ways:

Abstract

Please modify the headlines based on the format of the journal

-In the method section, please include the year of the study, sampling method, data analysis method.

Introduction

In this section, please bring this item

1- Definition of the research problem

2- The magnitude and importance of the study variable

3- Bringing the variable epidemiology information of the study at the world level, the country of India and the place of study

4- Bringing theoretical knowledge about the importance of studying

5- Expressing the necessity of conducting the study

Finally, the practical purpose of the study should be stated

Discussion

In the discussion section, it is necessary to compare the main results of the study with the results of other studies in this field,

To strengthen the article, especially in the introduction and discussion section the following studies are suggested, please to use and cited and add to article reference

- Post-traumatic stress disorder in medical workers involved in earthquake response: A systematic review and meta-analysis

- Anxiety, stress and depression levels among nurses of educational hospitals in Iran: Time of performing nursing care for suspected and confirmed COVID-19 patients

- Prevalence of workplace violence against health care workers in hospital and pre-hospital settings: An umbrella review of meta-analyses

Conclusion

What are your suggestion for future studies?

Best regards

6. PLOS authors have the option to publish the peer review history of their article (what does this mean?). If published, this will include your full peer review and any attached files.

Reviewer #1: No

Reviewer #2: No

---

## [Author Response · Author response to Decision Letter 0]

28 Jan 2024

Dear PLOS ONE editorial team,

Thank you for giving us the opportunity to submit a revised draft of the manuscript, and we would also like to thank you for your crucial comments on our paper (Manuscript ID: PONE-D-23-40921). We are very concerned and have combined all the suggested comments provided, which we believe strengthen our paper, and we hope this will render our paper eligible for consideration for publication in your reputed journal. We appreciate the time and effort that you and the reviewers dedicated to providing feedback on our manuscript and are grateful for the insightful comments and valuable improvements to our paper.

The authors would like to inform you that we have addressed the comments and recommendations made by both reviewers and the editor, point by point. In addition, throughout our revision, we made our best corrections too. All changes made to the original version are highlighted using tracking changes and attached as “Revised Manuscript with Track Changes." The unmarked copy of the manuscript is also attached as “Manuscript.” In addition, please see below a rebuttal letter that responds to each point raised by the academic editor and reviewers, and this letter is also attached to the submission as “Response to Reviewers."

---

## [Decision Letter · Decision Letter 1]

13 Feb 2024

PONE-D-23-40921R1Post-traumatic stress disorder and associated factors among internally displaced persons in Africa: A systematic review and meta-analysisPLOS ONE

Dear Dr. Tesfaye,

Thank you for submitting your manuscript to PLOS ONE. After careful consideration, we feel that it has merit but does not fully meet PLOS ONE’s publication criteria as it currently stands. Therefore, we invite you to submit a revised version of the manuscript that addresses the points raised during the review process.

Comments from the Handling Editor:

PTSD and PTS symptoms are two different things, and I understand that there can be strong limitations in the field to assess actual PTSD, and most of times only screenings can be done.

In order to avoid this potential bias in the global estimates, I recommend including a table with information regarding the tool used by the original studies for assessing PTSD. In this sense, did the original studies assess PTSD with clinical confirmation? Or did they assess screening of PTS symptoms without confirmation? Or mix of both assessments?

I also recommend a subgroup MA including clinically confirmed cases and positive screening cases as different sub-groups.

We look forward to receiving your revised manuscript.

Kind regards,

Roberto Ariel Abeldaño Zuñiga

Academic Editor

PLOS ONE

Additional Editor Comments:

PTSD and PTS symptoms are two different things, and I understand that there can be strong limitations in the field to assess actual PTSD, and most of times only screenings can be done.

In order to avoid this potential bias in the global estimates, I recommend including a table with information regarding the tool used by the original studies for assessing PTSD. In this sense, did the original studies assess PTSD with clinical confirmation? Or did they assess screening of PTS symptoms without confirmation? Or mix of both assessments?

I also recommend a subgroup MA including clinically confirmed cases and positive screening cases as different sub-groups.

Reviewers' comments:

Reviewer's Responses to Questions

**Comments to the Author**

1. If the authors have adequately addressed your comments raised in a previous round of review and you feel that this manuscript is now acceptable for publication, you may indicate that here to bypass the “Comments to the Author” section, enter your conflict of interest statement in the “Confidential to Editor” section, and submit your "Accept" recommendation.

Reviewer #1: All comments have been addressed

Reviewer #2: (No Response)

2. Is the manuscript technically sound, and do the data support the conclusions?

Reviewer #1: Yes

Reviewer #2: (No Response)

3. Has the statistical analysis been performed appropriately and rigorously? 

Reviewer #1: Yes

Reviewer #2: (No Response)

4. Have the authors made all data underlying the findings in their manuscript fully available?

Reviewer #1: Yes

Reviewer #2: (No Response)

5. Is the manuscript presented in an intelligible fashion and written in standard English?

Reviewer #1: Yes

Reviewer #2: (No Response)

6. Review Comments to the Author

Reviewer #1: (No Response)

Reviewer #2: Dear Authors

your performing systematic review and meta-analysis

32 to assess the prevalence and associated factors of PTSD among IDPs in Africa.

many thanks for your response.

Best regards

7. PLOS authors have the option to publish the peer review history of their article (what does this mean?). If published, this will include your full peer review and any attached files.

Reviewer #1: No

Reviewer #2: No

---

## [Author Response · Author response to Decision Letter 1]

3 Mar 2024

Dear PLOS ONE editorial team,

Thank you for giving us the opportunity to submit a revised draft of the manuscript, and we would also like to thank you for your crucial comments on our paper (Manuscript ID: PONE-D-23-40921R1). We are very concerned and have combined all the suggested comments provided, which we believe strengthen our paper, and we hope this will render our paper eligible for consideration for publication in your reputed journal. We appreciate the time and effort that you and the reviewers dedicated to providing feedback on our manuscript and are grateful for the insightful comments and valuable improvements to our paper for publication.

The authors would like to inform you that we have addressed the comments and recommendations of the handling editor point by point. In addition, throughout our revision, we made our best corrections too. All changes made to the original version are highlighted using tracking changes and attached as 'Revised Manuscript with Track Changes'. The unmarked copy of the manuscript is also attached as 'Manuscript'. In addition, please see below a rebuttal letter that responds to each point raised by the handling editor, and this letter is also attached to the submission as 'Response to Reviewers'.

We are very grateful to the editor and the two reviewers for their time and effort in improving our manuscript for publication. We thank them for their scientific comments.

---

## [Editor Report · Decision Letter 2]

6 Mar 2024

Post-traumatic stress disorder and associated factors among internally displaced persons in Africa: A systematic review and meta-analysis

PONE-D-23-40921R2

Dear Dr. Tesfaye,

We’re pleased to inform you that your manuscript has been judged scientifically suitable for publication and will be formally accepted for publication once it meets all outstanding technical requirements.

Kind regards,

Roberto Ariel Abeldaño Zuñiga

Academic Editor

PLOS ONE
---

## [Editor Report · Acceptance letter]

11 Mar 2024

PONE-D-23-40921R2 

PLOS ONE

Dear Dr. Tesfaye, 

I'm pleased to inform you that your manuscript has been deemed suitable for publication in PLOS ONE. Congratulations! Your manuscript is now being handed over to our production team.

Kind regards, 

on behalf of

Dr. Roberto Ariel Abeldaño Zuñiga 

Academic Editor

PLOS ONE